# Biological Control of Diamondback Moth—Increased Efficacy with Mixtures of *Beauveria* Fungi

**DOI:** 10.3390/microorganisms10030646

**Published:** 2022-03-17

**Authors:** Sereyboth Soth, Travis R. Glare, John G. Hampton, Stuart D. Card, Jenny J. Brookes

**Affiliations:** 1Bio-Protection Research Centre, Lincoln University, P.O. Box 85084, Lincoln 7647, New Zealand; travis.glare@lincoln.ac.nz (T.R.G.); john.hampton@lincoln.ac.nz (J.G.H.); jenny.brookes@lincoln.ac.nz (J.J.B.); 2Resilient Agriculture, AgResearch Limited, Grasslands Research Centre, Private Bag 11008, Palmerston North 4442, New Zealand; stuart.card@agresearch.co.nz

**Keywords:** biopesticide, biocontrol combinations, medium lethal concentration (LC50), medium lethal time (LT50), virulence

## Abstract

Diamondback moth (DBM) is an important horticultural pest worldwide as the larvae of these moths feed on the leaves of cruciferous vegetables. As DBM has developed resistance to more than 100 classes of synthetic insecticides, new biological control options are urgently required. *Beauveria* species are entomopathogenic fungi recognized as the most important fungal genus for controlling a wide range of agricultural, forestry, and veterinary arthropod pests. Previous research, aimed at developing new *Beauveria*-based biopesticides for DBM, has focused on screening single isolates of *Beauveria bassiana*. However, these fungal isolates have individual requirements, which may limit their effectiveness in some environments. This current study separately assessed 14 *Beauveria* isolates, from a range of habitats and aligned to four different species (*Beauveria bassiana, B. caledonica, B. malawiensis,* and *B. pseudobassiana*), to determine the most effective isolate for the control of DBM. Further assays then assessed whether selected combinations of these fungal isolates could increase the overall efficacy against DBM. Six *Beauveria* isolates (three *B. bassiana* and three *B. pseudobassiana*) achieved high DBM mortality at a low application rate with the first documented report of *B. pseudobassiana* able to kill 100% of DBM larvae. Further research determined that applications of low-virulent *Beauveria* isolates improved the control of DBM compared to mixtures containing high-virulent isolates. This novel approach increased the DBM pest mortality and shortened the time to kill.

## 1. Introduction

Brassica vegetables are cultivated in more than 150 countries worldwide with a combined value of around USD 40 billion [1]. In New Zealand, brassica vegetable production reaches around 115,700 metric tons per annum with a gross value of approximately USD 57 million [2]. Brassicas are susceptible to damage by many insect pests that can significantly reduce crop yield. One of the most destructive and predominant pests is diamondback moth (DBM), *Plutella xylostella*, which causes around USD 4–5 billion a year worldwide in brassica crop losses and control expenses [3]. Furthermore, this pest is the first recorded arthropod to develop resistance to more than 100 classes of synthetic insecticide, as well as the Bt toxin from the most common biopesticide, *Bacillus thuringiensis* [4,5,6,7] leaving very few control options available.

Biological control is an environmentally sound and effective means of reducing, or mitigating, insect pests in agriculture, aquatic, forest, natural resource, stored products, and urban environments (read Eilenberg et al. [8] for a detailed discussion). Biological control of insect pests is often achieved by the artificial introduction of free-living predators, parasitoids, or antagonistic microorganisms into a selected environment. *Beauveria* spp. are entomopathogenic fungi recognized as the most important fungal genus for controlling a wide range of agricultural, forestry, and veterinary arthropod pests [9]. *Beauveria*-based products are forecasted to reach a value of around USD 780 million by 2028 [10]. Species of *Beauveria* exhibit multiple mechanisms of biological control, including direct host cuticle penetration and the production of numerous bioactive secondary metabolites, which can limit the development of invertebrate pest resistance [9,11,12]. Insect infection with entomopathogenic fungi can also increase the adaptation capacity of certain insect species, decreasing their sensitivity to chemical insecticides [13]. *Beauveria* spp. are therefore ideal candidates for DBM control. Several studies have assessed *Beauveria bassiana* for the control of DBM with varying degrees of success [14,15,16,17,18,19]. However, these studies have focused on just one species of *Beauveria*, namely *B. bassiana*, and only assessed individual isolates at a time. There are more species in the genus that are potentially effective, but these have not been assessed. These less researched *Beauveria* species could have beneficial attributes that could improve biocontrol ability in the field.

Biological control agents aimed at controlling insect pests are vulnerable to environmental variability, fundamentally resulting in inconsistent field performance [20]. One of the ways to overcome this inconsistency and improve the environmental spectrum of bioactivity is to combine biocontrol agents and apply them as a mixture, an approach that is still rarely investigated. Combinations of genetically similar entomopathogenic fungi showed no improvement in overall virulence compared to the use of single fungal strains, but mixtures of genetically distinct strains led to improved antagonism toward the coffee borer beetle, and even synergism between fungal strains [21]. Combining mixtures of compatible biocontrol bacteria and entomopathogenic fungi can provide further improved efficacy by facilitating the combination of various traits. For example, combinations of plant-promoting rhizobacteria and entomopathogenic fungi reduced the incidence of leafminer and collar rot disease [22], while a combination of *B. thuringiensis* and *B. bassiana* provided a higher mortality of DBM than separate applications of the individual organisms [21]. Therefore, combining several genetically distinct isolates, which can also interact synergistically or overcome environmental constraints, may solve the inconsistent field performance issue.

The objectives of this study were to individually evaluate the virulence of 14 isolates of *Beauveria*, including several species, toward DBM, to determine the most efficacious isolates. The second objective was to evaluate combinations of *Beauveria* isolates, for any synergistic interactions.

## 2. Materials and Methods

### 2.1. Source of Beauveria Isolates and Maintenance

Fourteen isolates of *Beauveria*, sourced from two New Zealand collections, were used in this study (for more details, please see Table 1 in the Results section). Fungal colonies were purified using the single-spore isolation technique [23]. All isolates were maintained as axenic cultures on PDA (Oxoid) at 22 °C until required. For long-term storage, fungal hyphae were frozen within a 20% glycerol solution and stored at −80 °C.

### 2.2. DNA Extraction, PCR, and Sequencing 

Genomic DNA was extracted using the Chelex-100 method [24]. The elongation factor EF1-α primers (forward) 5′ (CARGAYGTBTACAAGATYGGTGG) and (reverse) 3′ (CCRAACRGCRACRGTYYGTCTCAT), and nuclear intergenic BLOC regions primers (forward) 5′ (GTC GCA GCC AGA GCA ACT) and (reverse) 3′ (AGA TTC GCA ACG TCA ACT T) were targeted for PCR amplification and sequencing. PCR amplification was performed in a total volume of 25 μL including 2 μL of genomic DNA extract, 2 μL (2.5 mM) of dNTPs, 1 μL (10 mM) of each primer (Integrated DNA Technologies Inc., Singapore), 2.5 μL of reaction buffer with MgCl_2_ (2 mM), 0.25 μL of Taq (0.7 U/reaction) (Roche Diagnostics GmbH, Mannheim, Germany), 0.5 μL of BSA, and 15.75 μL of PCR water. Amplification was conducted in a PCR thermal cycler (Kyratech Pty Limited, Brisbane, Australia) under 1 cycle of denaturation for 5 min at 95 °C, 1 cycle of run for 45 s at 95 °C, 40 cycles of annealing for 45 s at 57 °C, extending for 2 min at 72 °C, and concluding with 7 min of incubation at 72 °C. After the target amplification, PCR products were cleaned using the NucleoSpin PCR clean-up column (Macherey-Nagel GmbH & Co. KG, Düren, Germany) [25]. Then, all PCR products were sequenced using Sanger sequencing with the MagBio HighPrep (Thermo Fisher Scientific, Singapore) dye terminator removal (DTR) Clean-up protocol at the Bio-Protection Research Centre Lincoln University (Christchurch, New Zealand) sequencing facility, using a Hitachi ABI Prism 3130xl Genetic Analyser (Applied Biosystems, Singapore) with a 16-capillary 50 cm array installed, and using Performance-Optimized Polymer 7 (POP7) (Thermo Fisher Scientific, Singapore).

### 2.3. Phylogenetic Analysis 

To determine the species identity of the *Beauveria* isolates sourced from the New Zealand collections, sequences of these 14 isolates were compared with the sequences of a further 15 *Beauveria* isolates previously studied utilizing both EF1-α and BLOC sequence regions (Table 1). Sequence data were initially imported into the software package Geneious Prime^®^ 2021.0.1 (Biomatters Limited, Auckland, New Zealand) [26]. The forward and reverse directions from the same sequence were assembled using the de novo assembly to combine contigs into larger consensus reads [27], where regions of the poor match were trimmed using the Trim ends function before extracting the consensus files. The consensus result of each isolate was first confirmed using Nucleotide Basic Local Alignment Search Tool (BLASTN) [28]. After blasting using BLASTN (https://blast.ncbi.nlm.nih.gov/Blast.cgi?PAGE_TYPE=BlastSearch, accessed on 15 May 2021), all consensus data from EF1-α and BLOC were then aligned using the software package Clustal [29]. After aligning, longer sequences were manually trimmed to adjust the length of the shortest sequence. Trimmed results were then joined manually for EF1-α and BLOC and then extracted as combined consensus data. Additional comparison sequences were obtained from a Peruvian *Beauveria* isolate, UTRF21, phylogenetically similar to *B. bassiana* [30], and a New Zealand-derived isolate K4 [31], through the National Centre for Biotechnology Information (NCBI) website (https://www.ncbi.nlm.nih.gov/, accessed on 15 May 2021). Only isolates of *Beauveria* spp. within the NCBI database that had the EF1-α and BLOC regions sequenced were downloaded for use as reference isolates (Table 1). The sequences from the 15 additional isolates were also manually combined and extracted as combined consensus data. The sequence data from all 29 *Beauveria* isolates were multiple-aligned using the MUSCLE alignment software, the distance was measured using khmer4_6, clustering was performed using the Neighbor-Joining method, and tree rooting was performed using the pseudo method [32]. The sequences downloaded from NCBI were then trimmed manually to adjust the length of the sequence from the New Zealand-derived isolates in this study. The consensus trees were built from the combined sequence alignment by Neighbor-Joining as a nucleotide substitution [33] set to the Jukes–Cantor distance model [34] with no outgroup, with 1000 replications of Bootstrap and a support threshold of 50% in Geneious Prime^®^ tree builder [35].

### 2.4. Insect Bioassay 

Third-instar diamondback moth (DBM) larvae were obtained from a laboratory colony maintained at 25 ± 1 °C, 12:12 D/L, and fed on cabbage cv. Arisos NS, provided by South Pacific Seeds (Pukekohe Hill, Auckland, New Zealand) Limited, at the Bio-Protection Research Centre, Lincoln University, New Zealand.

Spores from the 14 *Beauveria* isolates obtained in New Zealand were harvested by adding 4 mL of an aqueous solution of 0.01% TritonTM X −100 to each Petri plate containing the fungal culture and gently brushing the colony with a sterile loop. The resulting crude suspension was filtered using cheese cloth to remove hyphal fragments and the conidial concentrations were estimated using a Neuman Bayer haemocytometer (Marienfeld, Darmstadt, Germany). Final spore concentrations were adjusted to 105, 107, and 108 conidia/mL for each *Beauveria* isolate and immediately placed at 4 °C until required. Spore suspensions were used within 12 h from harvest, although, at this temperature, *Beauveria* spores can retain their virulence for up to six months [36]. As the spray was only 600 µL, the application rates were 6 × 10^4^, 6 × 10^6^, and 6 × 10^7^ conidia/spray, and herein, these rates are referred to as low, medium, and high application rates, respectively.

Six-week-old cabbage leaves cv. Arisos NS, provided by South Pacific Seeds (Pukekohe, New Zealand) Limited, were collected from plants growing in a glasshouse and washed with 0.01% TritonTM X-100 (Sigma-Aldrich, Darmstadt, Germany) before being placed within a Petri plate containing water agar (WA, 12 g/L agar). Five third-instar DBM larvae were transferred to the surface of each leaf, two Petri plates per treatment, and replicated three times (six Petri plates with a total of 30 larvae). The Petri plates were covered with custom-made breathable lids (four holes were made in a lid using a hot metal cork borer; the lid was then covered with 1 mm × 1 mm fiber netting that was glued to the lid surface). A Potter Spray Tower (Burkard Manufacturing Co. Limited, Rickmansworth, UK) was used for inoculating DBM larvae with the *Beauveria* spore suspensions. Before spray inoculation, the liquid reservoir tube of the tower was cleaned with 1.2 mL of 70% ethanol and rinsed twice with 1.2 mL of 0.01% Triton X-100 and left to dry. This cleaning procedure was repeated after spraying each isolate. A 600 µL suspension of each *Beauveria* isolate, at each of the three prepared conidial concentrations, was loaded into the spray tower and applied to the Petri dishes containing the cabbage leaves and DBM larvae. The larvae were only sprayed once with each *Beauveria* isolate. The control consisted of Petri plates inoculated with 600 µL of an aqueous Triton X-100 solution (0.01%). The lids were placed on the Petri plates and these were subsequently randomized and placed on plastic trays. Trays were lined with water-soaked paper towels and covered by a plastic bag for 24 h to maintain high humidity. Petri plates were incubated at 23 ± 1 °C (12:12 D/L). Larval mortality was assessed daily under an Olympus SZX12 stereomicroscope for seven days (Shinjuku, Tokyo, Japan). A larva was considered dead when it did not respond when touched and had changed color from green to pale yellow or brown. To confirm that larvae had been killed by the fungal isolates, mycosed DBM cadavers were placed on glass microscope slides that included a thin layer of WA and were observed after 24 h of incubation (at 23 °C under 12:12 D/L), to assess the fungal sporulation characteristic of *Beauveria* spp.

### 2.5. Co-Inoculum of Selected Beauveria Isolates

Previous research had indicated that selected *Beauveria* isolates within our collection exhibited little to no antagonism toward each other when grown in dual culture [37]. This criterion was used to select *Beauveria* isolates for co-inoculation studies. Conidial suspensions, at three conidial concentrations, were prepared as described above for selected *Beauveria* isolates. Concentrations used were selected on the basis of the single isolate bioassays, to replicate low- to high-virulence challenges. Four co-inoculated groups were assigned: group A (a combination of isolates TPP-H, FRh2, and F532), group B (a combination of isolates F615, J2, and J18), group C (a combination of isolates FW Mana, I12 Damo, and CTL20), and group D (a combination of isolates F615, I12 Damo, and F532). Between each isolate, four controls were applied. The co-inoculant was then prepared by combining the conidial suspensions from three selected *Beauveria* isolates by pipetting 1 mL of each suspension into a fresh centrifuge tube. Only suspensions of the same conidial concentration were combined. A 600 µL suspension of the co-inoculated isolates was applied to DBM larvae and their mortality assessed as described earlier.

### 2.6. Statistical Analysis

Probit analysis was used to evaluate the DBM larval mortality data (LC50) [38]. The natural mortality of DBM larvae was determined by using the control treatment (0.01% Triton X-100) with less than 20% mortality acceptable [39]; otherwise, the bioassay was repeated until this limit was achieved. Abbott’s formula was used to correct for natural DBM larval mortality [40]. For multiple comparisons of LC50, each replicate was analyzed separately before combining them for ANOVA analysis. Tukey’s honestly significant difference (HSD) test was used to compare individual treatment means [41]. The medium lethal time (LT50) was used to test for the speed of kill of each isolate. The formula LT50 = [ND50 × (M50 − 1) + DB × (NM50 − M50)]/(NM50 − 1), where ND50: number of days that mortality reached 50%; M50: 50% mortality of tested larvae; DB: the day started to die before 50% mortality; and NM50: number of total mortality on ND50 (personal communication, Dave Saville, 20 April 2021). For *Beauveria* isolates that achieved lower than 50% DBM larval mortality within the experimental time, an LT50 could not be calculated using this formula. The formula LT50 prediction = (M50 × ND)/MD, where M50: 50% mortality of tested larvae; ND: number of days tested; and MD: total mortality of days tested was therefore used. Statistical analysis was conducted using Genstat 20th Edition [42].

## 3. Results

### 3.1. Phylogenetic Comparisons

Consensus data for the sequences consisted of 539 and 353 bp for EF1-α and BLOC, respectively. After deleting gaps and ambiguous alignments, 527 and 308 bp of EF1-α and BLOC sequences remained, respectively. A phylogenetic tree was produced using the combined-sequence data for EF1-α and BLOC, comparing the sequences of the 14 *Beauveria* isolates obtained in New Zealand to 15 reference isolates. The phylogenetic tree confirmed that the 14 *Beauveria* isolates from New Zealand collections were aligned with four species: *B. bassiana*, *B. caledonica*, *B. malawiensis*, and *B. pseudobassiana* (Table 1 and Figure 1). The sequences of isolates CTA20, CTL20, J2, J18, and Mo1 were identical, aligning with reference isolate K4 of *B. bassiana*. This clade was phylogenetically similar to O2380. TPP-H was closely aligned with *B. bassiana* reference isolate ARSEF1040, while isolates FRh2 and F615 were aligned with the reference isolates ARSEF7518 and ARSEF1564 (Figure 1). The sequences of isolates F532 and FRh1 were identical and aligned closely with the *B. caledonica* reference isolate ARSEF8024, while isolate Bweta was identical to the *B. malawiensis* reference isolate ARSEF4755 (Figure 1). The isolates FRhp, I12 Damo, and FW Mana aligned closely with the reference *B. pseudobassiana* isolates ARSEF1855 and ARSEF3529 (Figure 1).

**Table 1 microorganisms-10-00646-t001:** A list of the *Beauveria* isolates including (1) those obtained from New Zealand collections assessed for the control of diamondback moth (DBM) and (2) those used for comparative purposes in the phylogenetic study.

Species	Country of Origin	Isolate Code	Host or Habitat	BLOC Sequence	EF1-α Sequence	Used in DBM Bioassays or Only for Phylogenetic Analysis ^1^
*Beauveria amorpha*	Australia	ARSEF4149	Coleoptera: Scarabaeidae	HQ880735	HQ881006	Phylogenetic analysis [43]
*B. australis*	Australia	ARSEF4580	Orthoptera: Acrididae	HQ880719	HQ880994	Phylogenetic analysis [43]
*B. bassiana*	Japan Italy Japan New Zealand New Zealand New Zealand New Zealand New Zealand New Zealand New Zealand New Zealand USA New Zealand	ARSEF1040 ARSEF1564 ARSEF7518 K4 CTA20 CTL20 F615F Rh2 J2 J18 Mo1 O2380 TPP-H	Lepidoptera: Bombycidae Orthoptera: Arctiidae Hymenoptera: Pamphiliidae Coleoptera: Scolytidae Lepidoptera: Plutellidae Lepidoptera: Plutellidae Organic soil Coleoptera: Scolytidae Lepidoptera: Plutellidae *Zea mays* Lepidoptera: Plutellidae *Brassica rapa* Hemiptera: Triozidae	HQ880689 HQ880692 HQ880693 MW030951 MZ703290 MZ703289 MZ703288 MZ703287 MZ703286 MZ703285 MZ703284 MZ703283 MZ703282	AY531881 HQ880974 HQ880975 MW030949 MZ703304 MZ703303 MZ703302 MZ703301 MZ703300 MZ703299 MZ703298 MZ703297 MZ703296	Phylogenetic analysis [43] Phylogenetic analysis [43] Phylogenetic analysis [43] Phylogenetic analysis [31] DBM bioassay (BPRC) DBM bioassay (BPRC) DBM bioassay (BPRC) DBM bioassay (BPRC) DBM bioassay (BPRC) Phylogenetic analysis [44] DBM bioassay (BPRC) DBM bioassay (AgResearch) DBM bioassay (BPRC)
*B. brongniartii*	Japan	ARSEF7516	Coleoptera: Scarabaeidae	HQ880697	HQ880976	Phylogenetic analysis [43]
*B. caledonica*	New Zealand Denmark New Zealand	FRh1 ARSEF8024 F532	Coleoptera: Scolytidae Coleoptera: Scarabaeidae Coleoptera: Scolytidae	MW030952 HQ880749 MZ703281	MW030947 HQ881012 MZ703295	Phylogenetic analysis [31] Phylogenetic analysis [43] DBM bioassay (BPRC)
*B. kipukae*	USA	ARSEF7032	Homoptera: Delphacidae	HQ880734	HQ881005	Phylogenetic analysis [43]
*B. malawiensis*	Australia New Zealand New Zealand	ARSEF4755 Bweta* Bweta	Soil Orthoptera: Anostostomatidae Orthoptera: Anostostomatidae	HQ880754 MW030953 MZ703280	HQ881015 MW030946 MZ703294	Phylogenetic analysis [43] Phylogenetic analysis [31] DBM bioassay (BPRC)
*B. peruviensis*	Peru	UTRF21	Coleoptera: Curculionidae	MN094752	MN094767	Phylogenetic analysis [30]
*B. pseudobassiana*	USA Canada New Zealand New Zealand New Zealand	ARSEF3529 ARSEF1855 FRhp FW ManaI12 Damo	Lepidoptera: Lymantriidae Coleoptera: Scolytidae Laboratory contaminant Coleoptera: Curculionidae Coleoptera: Coccinellidae	HQ880726 HQ880727 MZ703279 MZ703278 MZ703277	HQ880998 HQ880999 MZ703293 MZ703292 MZ703291	Phylogenetic analysis [44] Phylogenetic analysis [44] DBM bioassay (BPRC) DBM bioassay (BPRC) DBM bioassay (BPRC)

^1^ If used in phylogenetic analysis, a reference is also provided, or if only used for insect bioassays, the original collection is listed (BPRC = Bio-Protection Research Centre, Lincoln University, New Zealand). Bweta*: the same isolate but EF1-α only used for phylogenetic analysis.

### 3.2. Single Inoculum of Beauveria Isolates 

#### 3.2.1. DBM Larval Mortality

All of the 14 *Beauveria* isolates assessed in this study were pathogenic toward DBM larvae, with the mortality rate ranging from 4 to 100% within seven days post-inoculation (Table 2). The most virulent isolates were *B. bassiana* Mo1, CTL20, CTA20; and *B. pseudobassiana* FRhp, FW Mana, and I12 Damo, which killed up to 100% of larvae within seven days at the high application rate (Table 2). Furthermore, isolates FRhp, FW Mana, I12 Damo, and CTL20 also killed 100% of DBM larvae at the medium application rate (Table 2). The most virulent isolates were I12 Damo and CTL20, which resulted in a mean larval mortality of 46% at the lower application rate, while the less virulent isolates were *B. malawiensis* Bweta and *B. caledonica* F532 that killed less than 40% at the high application rate (Table 2).

#### 3.2.2. Medium Lethal Concentration (LC50) and Medium Lethal Time (LT50)

The LC50 values differed significantly (F2, 13 = 11.66, *p* < 0.001) among the 14 assessed *Beauveria* isolates (Figure 2). The most virulent isolates, Mo1, FRhp, I12 Damo, FW Mana, CTA20, and CTL20, required the lowest application rate of 10^5^ conidia/spray to achieve the LC50, while isolates F615, J2, J18, and O2380 required an application rate of 10^6^ conidia/spray to achieve the LC50. The least virulent isolates that required the higher application rate to reach the LC50 were TTP-H and FRh2, requiring 10^7^ conidia/spray, and F532 and Bweta, requiring the highest application rate of 10^8^ conidia/spray to achieve the LC50 (Figure 2).

No significant variation (F2, 13 = 2.10, *p* = 0.053) was found among the LT50 estimates of all *Beauveria* isolates tested at the low application rate (Figure 3). However, there was a significant difference in the LT50 values at the medium application rate (F2, 13 = 2.33, *p* = 0.032).

There were further significant interactions between *Beauveria* isolates at the highest rate of application (F2, 13 = 2.68, *p* = 0.016). Isolates J2, Mo1, FRhp, I12 Damo, CTL20, and CTA20 required 2–3 days to achieve the LT50, while most of the remaining isolates (with the exception of Bweta) required 4–6 days to reach the LT50 (Figure 3).

#### 3.2.3. Percentage of DBM Larval Cadavers Supporting *Beauveria* Sporulation

The fungal sporulation on cadavers was concentration-dependent. At the low application rate, only isolates I12 Damo, CTA20, and F532 provided 100% of cadavers supporting sporulation, followed by isolates FRhp, Mo1, and CTL20, which differed from other isolates (F2, 13 = 50.58, *p* < 0.001). At the medium application rate, isolates I12 Damo, CTA20, FRhp, and Mo1 resulted in 100% sporulation, followed by CTL20, FW Mana, J18, Bweta, F615, and O2380, which were significantly higher than other isolates (F2, 13 = 14.19, *p* < 0.001). At the high application rate, only isolate J18 was significantly lower than other isolates (F2, 13 = 3.52, *p* < 0.003) (Figure 4).

### 3.3. Co-Inoculum of Beauveria Isolates on DBM 

#### 3.3.1. DBM Mortality

At the high application rate, the treatments that contained a co-inoculum of *Beauveria* isolates achieved >90% larval mortality of DBM (Table 3). Co-inoculum groups A, C and D, alongside isolates FW Mana, I12 Damo and CTL20 used singularly, achieved 100% larval mortality of DBM at the high application rate (Table 3). Co-inoculum group C, alongside isolates FW Mana, I12 Damo, and CTL20 used singularly, was the only co-inoculum group to also achieve 100% larval mortality of DBM at the medium application rate (Table 3). None of the treatments, either *Beauveria* isolates used within a co-inoculum or used singularly, achieved 100% larval mortality at the low application rate (Table 3). The highest performing treatments at the lowest rate of application were I12 Damo and CTL20, which both achieved 46% DBM larval mortality, although this was not significantly different (*p* > 0.05) to the other treatments at this application rate. Co-inoculum group A achieved a higher rate of DBM mortality than when TPP-H, FRh2, and F532 were used as singular applications at all the application rates assessed (Table 3). Co-inoculum group B achieved a higher rate of DBM mortality than when F615, J2, and J18 were applied singularly at the highest rate of application (Table 3). Co-inoculum group C achieved a similar rate of DBM mortality to when FW Mana, I12 Damo, and CTL20 were applied as singular applications at all the application rates (Table 3), while co-inoculum group D achieved a similar rate of DBM mortality to when F615, I12 Damo, and F532 were used as singular applications at all the application rates (Table 3).

#### 3.3.2. Medium Lethal Dose (LC50) and Medium Lethal Time (LT50)

LC50 for the four combination treatments did not differ from that of any single isolate with the exception for FRh2 and F532 (Figure 5). Co-inoculum groups A, B, and C and the single isolate applications of I12 Damo and CTL20 required around 10^5^ conidia/spray to cause 50% mortality. Co-inoculum group D required roughly 10^6^ conidia/spray to achieve LC50. In comparison to the single isolate applications, co-inoculum group A, the less virulent combination, required lower inoculum rates. Co-inoculum group D did not differ from the medium- and lower-virulence isolates, FW Mana, F615, J18, and J2, which required around 3–8 × 10^6^ conidia/spray to achieve 50% mortality (Figure 5).

At the low application rate, the LT50 did not differ (F2, 12 = 1.57, *p* = 0.224) among the four co-inoculum treatments (Figure 6). The co-inoculum groups A and B reached LT50 at almost the same time as the individual isolates I12 Damo and CTL20 (Figure 6). All four co-inoculum groups took around four to five days to achieve LT50 at the medium application rate, which differed (F2, 12 = 8.45, *p* < 0.001) significantly. At the medium application rate, isolate FRh2 required a significantly longer time to achieve the LT50 compared to other isolates and co-inoculum groups (Figure 6). At the high application rate, all four co-inoculum groups were only significantly different (F2, 12 = 11.58, *p* < 0.001) from individual isolate *B. caledonica* F532 (Figure 6). At the high application rate, co-inoculum group C, however, was significantly different (F2, 12 = 11.58, *p* < 0.001) from individual isolates *B. bassiana* TPP-H, *B. bassiana* FRh2, and *B. caledonica* F532 (Figure 6).

#### 3.3.3. Percentage of Cadavers Supporting Sporulation

The percentage of cadavers that supported sporulation ranged from 42% to 95% for all four co-inoculum groups across the three application rates. Co-inoculum group C (the high-virulence combination) resulted in 95% of cadavers supporting sporulation at the high application rate followed by co-inoculum group B (the medium-virulence combination) (90%), co-inoculum group D (a combination of three species) (85%), and co-inoculum group A (the low-virulence combination) (75%) (Figure 7). At the medium application rate, co-inoculum group A resulted in a sporulation percentage of 78%, that of co-inoculum group C was 75%, and those of co-inoculum groups B and D were both 61% (Figure 7). However, at both application rates, the differences were not significant (Figure 7). At the low application rate, co-inoculum group C (the high-virulence combination) resulted in 100% sporulation, followed by co-inoculum group B (the medium-virulence combination) (87.5%), co-inoculum group D (the three species combination) (83%), and co-inoculum group A (the low-virulence combination) (42%) (Figure 7). There was a statistically significant difference between co-inoculum group C, the highly virulent, and co-inoculum group A, the less virulent combinations (F2, 3 = 12.66, *p* = 0.033) (Figure 7).

## 4. Discussion

This study showed that all 14 *Beauveria* isolates used in the study, selected from two New Zealand collections, were pathogenic to diamondback moth (DBM) larvae within an in vivo bioassay. The most virulent isolates were sequenced and aligned with two species of *Beauveria*, namely *B. bassiana* and *B. pseudobassiana.* Three isolates of *B. bassiana* (Mo1, CTL20, and CTA20) and three isolates of *B. pseudobassiana* (FRhp, FW Mana, and I12 Damo) killed 100% of DBM larvae within seven days at the high application rate. Previous studies around the world have also noted the high virulence of *B. bassiana* toward DBM [14,15,16,17,45,46,47], while studies using *B. pseudobassiana* have yet to be undertaken. The three highly virulent isolates of *B. bassiana* (Mo1, CTL20, and CTA20) were reisolated from cadavers that were likely infected by isolate J18 (personal communication, Jenny Brookes). The phylogenetic analysis also confirmed their identities, which are on the same branch of the phylogenetic tree. While isolate J18 was not highly virulent toward DBM larvae, this result leads to the hypothesis that introducing a fungal isolate to an insect colony enhances virulent ability. In a previous study, isolate *B. bassiana* 1200 that was reisolated from the DBM host killed 97% of DBM larvae, while the original isolate that was introduced to the insect colony killed only 87% [48]. Recently published studies have demonstrated that passage through a host insect can change certain entomopathogens’ methylation patterns, resulting in increased virulence [49,50]. For example, an isolate of *Metarhizium robertsii* lc-2575 exhibited higher virulence after the fungus had been experimentally passaged through an insect larva. Furthermore, the passage of this isolate through plant roots or insects resulted in increased conidiation [51]. Further work could aim to increase the virulence of selected isolates of *Beauveria* by passaging through DBM larvae.

While isolates of *B. pseudobassiana* have been found to be highly virulent to bark beetles, silkworms, and mealworms [52], our study is the first to report isolates of *B. pseudobassiana* able to kill 100% of DBM larvae. The three *B. pseudobassiana* isolates used in the current study were derived from coleopteran hosts from New Zealand habitats. Among them, isolate FW Mana was previously tested against flax weevil, resulting in high mortality [53]. There is limited information about isolates I12 Damo and FRhp as they are new to this study. The three *B. pseudobassiana* isolates evaluated in this study gave a high level of virulence toward DBM, a lepidopteran pest, even though they were all originally isolated from coleopteran hosts. Romón et al. [54] found a similar response for *B. pseudobassiana* except the reverse was found, in that isolates derived from a lepidopteran host were significantly more virulent to *Pissodes nemorensis*, a coleopteran pest, than isolates derived from a coleopteran host. Based on these findings, *B. pseudobassiana* isolates may provide a broader host range among insects than *B. bassiana*. This characteristic is beneficial for the development of new biopesticides as long as the host range does not include beneficial invertebrates [55,56]. For example, the failure of some effective *Beauveria* isolates was due to their narrow host control [20,57]. Therefore, using these three *B. pseudobassiana* isolates to test against other insect pests is required to examine their host range.

Our study also uncovered great diversity within this group of *Beauveria* isolates with respect to their bioactivity toward DBM, with *B. caledonica* isolate F532 and *B. malawiensis* isolate Bweta achieving a low mortality of DBM larvae at the rates of application evaluated. Reay et al. [58] also showed a low mortality with *B. caledonica* F532 against *Hylastes ater*(*Coleoptera*) and, therefore, this isolate may have a low level of virulence against many invertebrate species. However, the same study found high mortality of the tested insect using *B. malawiensis* isolates. Interestingly, isolate *B. caledonica* F532 was found to provide high mortality against *Hylurgus ligniperda* adults and *Tenebrio molitor* larvae [39]. The host specificity of isolates *B. caledonica* F532 and *B. malawiensis* Bweta may result in a low mortality of DBM.

Isolates CTL20 and I12 Damo required a low application rate to kill 95% of larvae. These two isolates required the same rates as reported for a commercially developed isolate GHA [46,47], isolate IBCB01 [16], and isolate ARSEF9271 [59], but lower rates than isolate MG-Bb-1 [15] and isolate ESALQ-447 [19]. The isolate *B. bassiana* GHA has been developed into several commercially available biopesticide products (BotaniGard^®^ ES and Mycotrol^®^) (BioWorks, Inc., New York, NY, USA) in several countries, including Australia, Canada, and the US [46,47,60]. The isolate is able to control a diversity of invertebrate pests, including DBM [48], Colorado potato beetle [61], emerald ash borer [62], tobacco whitefly [63], plant bugs [64], coffee borer beetle [65], house fly [66], western cherry fruit fly [67], and fall armyworm [68]. Testing isolates CTL20 and I12 Damo against other insect pests is required to assess their host range and examine their consistency. For killing time, isolates CTL20 and I12 Damo killed the target host faster than isolates used in other studies [48,69,70,71]. These findings revealed that *Beauveria* isolates of this study might have more potential to control particular pests than those isolates already developed or introduced from other locations. For example, isolate *B. bassiana* 1200, an indigenous isolate, also killed faster than isolate GHA [48]. As our experiments were conducted under a laboratory setting, further trials must be conducted to evaluate the efficacy of these two *Beauveria* isolates under greenhouse and field conditions. Some *Beauveria* isolates have been shown to be safe to beneficial insects [9,11]. Future research would also determine whether a broader range of targeted insect pests could be controlled by these isolates while exhibiting little to no bioactivity toward beneficial arthropods. Isolate CTL20 and I12 Damo showed three positive characteristics: (1) requiring a low application rate to kill DBM larvae, (2) fast to kill, and (3) growing fast on culturing medium (these two isolates produced up to 10^9^ conidia/plate within three weeks). These characteristics are crucial for the development of fungal-based mycopesticides [9,72]. For example, Mycotrol^®^ (BioWorks, Inc., New York, NY, USA), a product based on *B. bassiana* conidia is very successful due to the high spore production on solid substrate media [73]. 

This efficient colonization of the cadaver and sporulation is an indication of the strongly self-sustaining characteristic of these isolates. From a financial point of view, growers need biopesticides that can regenerate their mode of action to infect target pests for more than one generation [74]. As *Beauveria* can sporulate quickly after host infection, these fungi can replicate through horizontal transmission, which may lead to long-term suppression of the treated pest populations without a need to repeat the application [20,75]. The result of this study demonstrated the highly prolific sporulation of the most effective isolates. However, it is also important to look for appropriate transmission technologies to avoid any possible infections of beneficial insects. *Beauveria* spp. have generally proved to be safe for nontarget organisms [11,76,77]. Beaublast (Biotelliga Holdings Limited., Auckland, New Zealand), a commercial product based on *B. bassiana*, has been commercialized and used in New Zealand [78], establishing a pathway to registration. Thus, if the highly virulent isolates of this study prove to have no negative impacts on beneficial insects, they are the most suitable candidates for *Beauveria*-based biopesticides for DBM control in New Zealand.

The combination of isolates showed two separate scenarios: synergism and antagonism. Usually, combining different isolates results in cooperation to kill the host [79,80]. In this study, the best results were obtained from combining three less virulent isolates, although the mortality did not surpass the sum of these three isolates. However, the combination resulted in faster mortality than those of the individual isolates. The low-virulent combination caused 100% mortality within six days after spraying, while the individual isolates killed around 50% within seven days post-treatment. Cruz et al. [80] found that combining low-virulent isolates showed better results when testing against coffee berry borer than single isolates. Under field conditions, a commercial product that combines three *Beauveria* strains, Cenicafé, reduced the population of coffee berry borer emerging from infested beans by up to 75% [81], and when combining this commercial product with *Metarhizium anisopliae* Ma9236, the insect population declined by 93% [82]. These findings revealed the consistency of combined isolates under field conditions. The efficacy of our combined-isolate treatments may be the result of gene recombination through the sexual mating-type. For instance, when observing Masson pine caterpillar larvae, Wang et al. [83] proved that some *B. bassiana* isolates could breed or join together through sexual mating to infect the insect under natural conditions, which provided synergism. However, the teleomorph characteristic of *B. bassiana* was found to be a rare occurrence in nature, and it is even difficult to investigate in the laboratory [84]. When *Beauveria* spp. were described as *Cordyceps* spp., Yokoyama et al. [85] reported that mating-type genes (MAT1-1-1 and MAT1-2-1) were amplified better than the 18S rRNA gene using phylogenetic base analysis. Some sexual mating genes found in the fungi Aspergillus nidulans and Neurospora crassa were observed in B. bassiana [86]. The absence of the meiotic recombination Spo11-like protein means that sexual reproduction in Beauveria strains rarely takes place [87]. These reports showed that Beauveria spp. also have sexual mating characteristics, and the lack of Spo11 protein resulted in a lower frequency of this reproductive type. The LC50 and LT50 of the low-virulent combination were as low and quick as a highly virulent isolate such as I12 Damo and CTL20 and lower and faster than the individual isolates used in the combination. This result is aligned with the co-infection model [79].

An antagonistic interaction was found when mixing isolates of three species of *Beauveria*. This combination gave significantly lower mortality than the single species isolates at low application rates, except for isolate F532, which caused lower mortality singly anyway. Bayman et al. [88] found the same result when treating combinations of isolates on coffee borer beetle in Puerto Rico. The highly virulent strains may dominate the least virulent strains for food resources, reducing the speed of host infection when combining them in an application [80,89]. As isolate I12 Damo was highly infectious for DBM larvae, this isolate might dominate the others, infecting the host but retarding the speed of kill. For isolates that cause high mortality for a particular insect, applying a single application is likely the more effective approach [90]. Therefore, we assume that the application using only isolate I12 Damo will be enough to cause 100% mortality of DBM. The two other combinations provided no obvious interactions. The highly virulent combination caused the same mortality and speed of kill as individual isolates. Even though the medium-virulent combination killed relatively more than single isolates, there were no synergistic nor antagonistic results. However, these two combinations may work more consistently under field conditions compared to single isolates as genetic diversity might improve the range of fungal ability to tolerate varied environments [83].

There was a reduction in the percentage of infected cadavers sporulating in all combinations compared to in the single isolate bioassays. Combining different isolates can result in competition for food resources for invasion inside a host, resulting in killing faster but reducing sporulation [80,89]. This characteristic may be beneficial if the infective isolate was reisolated and used as individual treatment. For instance, the co-inoculum of *B. bassiana* and *B. brongniartii* revealed that different metabolites were produced compared to single isolates sprayed separately [91]. Thus, combined isolates may also produce metabolites to kill DBM larvae while killing by spores simultaneously. This may not be a negative effect for a commercial product, as the lack of sporulation would mean the product must be applied regularly, increasing sales.

## 5. Conclusions

This study found that three isolates of *B. bassiana* and three of *B. pseudobassiana* are potential candidates for the control of diamondback moth (DBM). The three isolates of *B. pseudobassiana* also caused 100% mortality. This is the first report to have shown that New Zealand *B. pseudobassiana* isolates are suitable candidates for DBM control. Combining three low virulent isolates caused a higher mortality with a faster kill speed than those of the highly virulent combinations and isolates. Combining three *Beauveria* species resulted in antagonism. For future prospects, experimental studies must involve evaluating suitable media for mass production, examining side-effects on beneficial insects, looking for safe delivery technologies, and integrating with the other control measures for integrated pest management for sustainable DBM management.

## Figures and Tables

**Figure 1 microorganisms-10-00646-f001:**
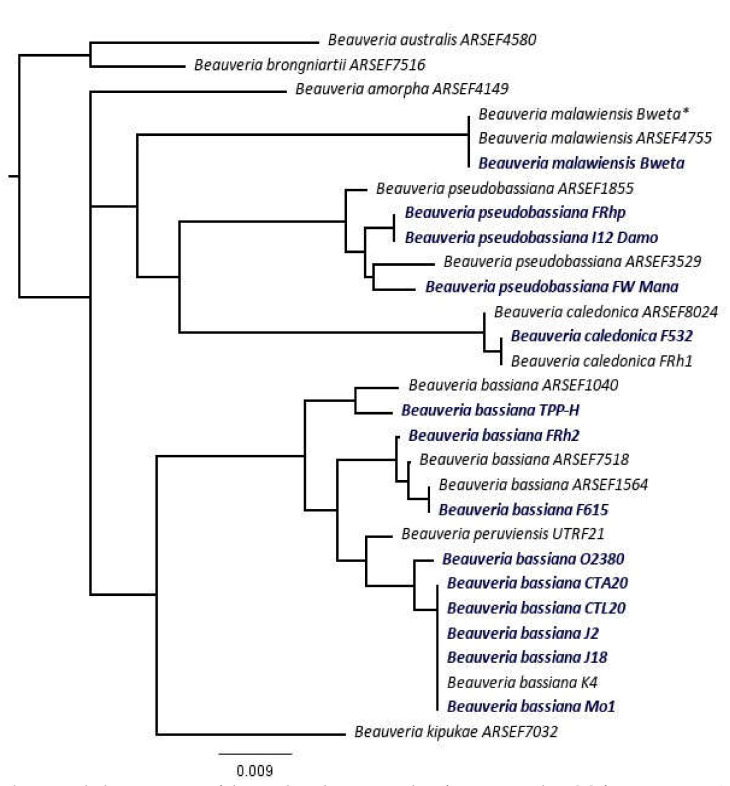
Phylogenetic tree of the combined-sequence data for EF1-α and BLOC from 14 *Beauveria* isolates derived from New Zealand collections (in bold) and 15 reference isolates based on the Neighbor-Joining method using Juke–Cantor distance model, with no outgroup and 1000 replicates of Bootstrapping. A support threshold of 50% was set in Geneious tree builder. Isolate Bweta appears twice in the figure, once as a reference isolate (Bweta*) and once for phylogenetic analysis using EF1-α + BLOC (bold).

**Figure 2 microorganisms-10-00646-f002:**
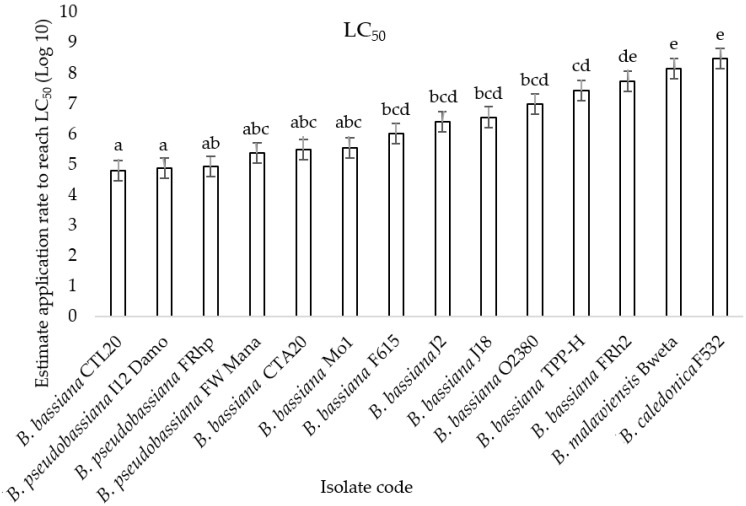
The medium lethal concentration (LC50) values (±SE) required to kill diamondback moth larvae for 14 selected isolates of *Beauveria*. Bars followed by the same letter are not significantly different according to Tukey’s HSD test (*p* > 0.05).

**Figure 3 microorganisms-10-00646-f003:**
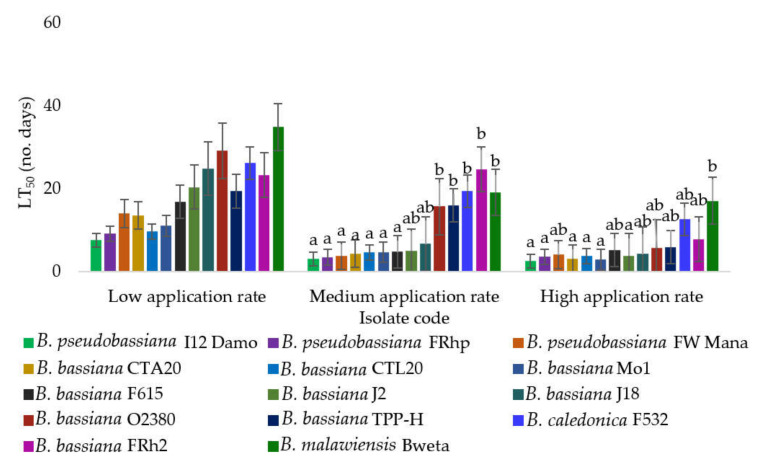
The LT50 values (±SE) after exposure of diamondback moth larvae to 14 selected isolates of *Beauveria*. Bars followed by the same letter are not significantly different according to Tukey’s HSD test.

**Figure 4 microorganisms-10-00646-f004:**
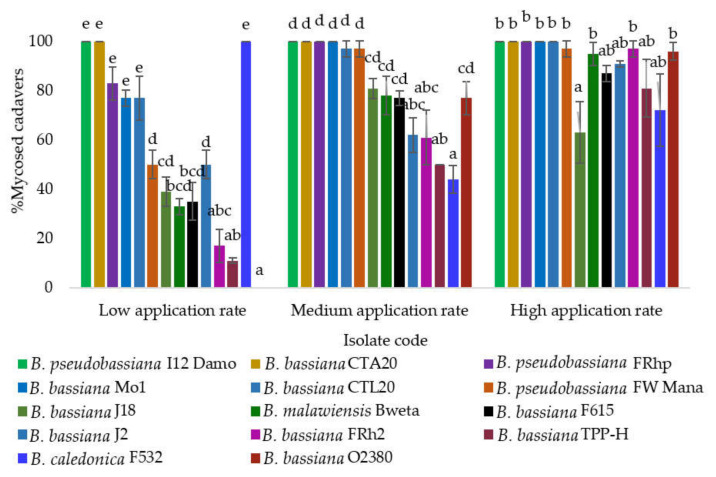
The percentage of diamondback moth larval cadavers that supported fungal sporulation after exposure to 14 selected isolates of *Beauveria* at three application rates (error bars following letters represent ANOVA, multiple comparisons using Tukey’s HSD test).

**Figure 5 microorganisms-10-00646-f005:**
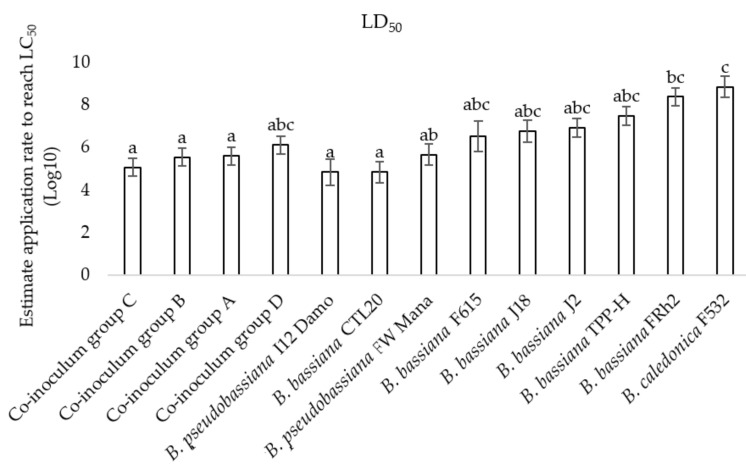
LC50 for the four combinations of isolates compared to the individual isolates (error bars following letters represent ANOVA; multiple comparisons using Tukey’s HSD test) (F2, 12 = 6.47, *p* = 0.001).

**Figure 6 microorganisms-10-00646-f006:**
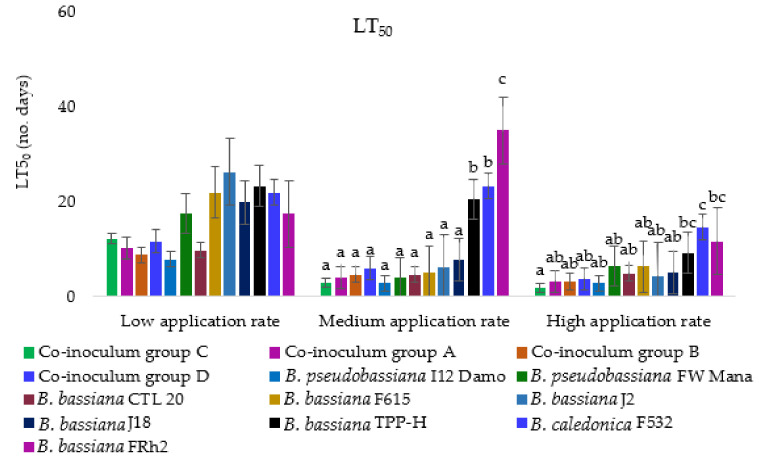
LT50 for the four combinations of isolates and the individual isolates (error bars following letters represent ANOVA; multiple comparisons using Tukey’s HSD test).

**Figure 7 microorganisms-10-00646-f007:**
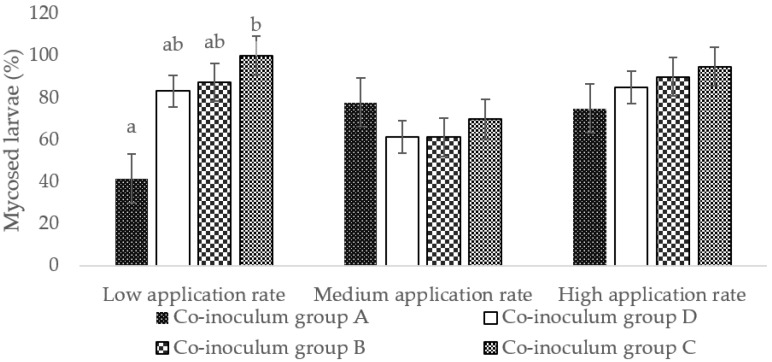
Percentage of cadavers supporting sporulation after the four combined treatments at the three application rates (error bars following letters represent ANOVA; multiple comparisons using Tukey’s HSD test).

**Table 2 microorganisms-10-00646-t002:** Larval mortality (% ± SE) of diamondback moth (DBM) seven days after application of selected *Beauveria* isolates at three conidial mortality rates was corrected against the control mortality rate using Abbott’s formula. Values in a column followed by the same letter are not significantly different (*p* > 0.05) according to Tukey’s HSD test.

Species and Isolate Code	Mean Larval Mortality of DBM at Different Conidial Application Rates (Conidia/Spray)		
	Low (6 × 10^4^)	Medium (6 × 10^6^)	High (6 × 10^7^)
*Beauveria pseudobassiana* I12 Damo	46 ± 2.87	100 ± 2.31 ^d^	100 ± 4.30 ^b^
*B. bassiana* CTL20	46 ± 1.60	100 ± 4.02 ^d^	100 ± 3.05 ^b^
*B. pseudobassiana* FRhp	36 ± 1.91	100 ± 4.19 ^d^	100 ± 4.08 ^b^
*B. pseudobassiana* FW Mana	25 ± 0.87	100 ± 3.76 ^d^	100 ± 4.46 ^b^
*B. bassiana* CTA20	25 ± 1.26	89 ± 4.76 ^d^	100 ± 3.84 ^b^
*B. bassiana* Mo1	36 ± 1.32	79 ± 3.73 ^cd^	100 ± 4.02 ^b^
*B. bassiana* F615	36 ± 1.46	89 ± 3.51 ^d^	79 ± 3.05 ^ab^
*B. bassiana* J2	25 ± 0.87	79 ± 2.37 ^cd^	79 ± 3.60 ^ab^
*B. bassiana* J18	25 ± 0.76	68 ± 2.37 ^bcd^	79 ± 2.90 ^ab^
*B. bassiana* TPP-H	14 ± 0.55	36 ± 1.57 ^ab^	57 ± 1.59 ^ab^
*B. bassiana* O2380	4 ± 0.41	57 ± 2.22 ^abc^	79 ± 3.64 ^ab^
*B. bassiana* FRh2	14 ± 0.37	14 ± 0.55 ^a^	68 ± 2.12 ^ab^
*B. malawiensis* Bweta	4 ± 0.03	25 ± 0.37 ^a^	36 ± 1.93 ^a^
*B. caledonica* F532	14 ± 0.28	25 ± 1.20 ^a^	25 ± 1.30 ^a^
F2, 13	1.15	10.54	6.01
*p*-value	0.364	<0.001	<0.001

**Table 3 microorganisms-10-00646-t003:** Larval mortality (% ± SE) of diamondback moth seven days after application with selected *Beauveria* isolates at three application rates. Mortality rates were corrected against the control mortality rate using Abbott’s formula. Values in a column followed by the same letter are not significantly different (*p* > 0.05) according to Tukey’s HSD test.

Isolate	Application Rate (Conidia/Spray)		
	Low (6 × 10^4^)	Medium (6 × 10^6^)	High (6 × 10^7^)
	Mean ± SE	Mean ± SE	Mean ± SE
Co-inoculum group A	28 ± 2.18	78 ± 3.89 ^de^	100 ± 8.41 ^b^
Co-inoculum group B	33 ± 2.77	83 ± 4.74 ^de^	94 ± 4.97 ^b^
Co-inoculum group C	28 ± 3.62	100 ± 3.85 ^e^	100 ± 8.34 ^b^
Co-inoculum group D	22 ± 2.30	56 ± 2.10 ^bcd^	100 ± 3.85 ^b^
*B. bassiana* F615	36 ± 1.46	89 ± 3.51 ^de^	79 ± 3.05 ^ab^
*B. bassiana* J2	25 ± 0.87	79 ± 2.37 ^cd^	79 ± 3.60 ^ab^
*B. bassiana* TPP-H	14 ± 0.55	36 ± 1.57 ^abc^	57 ± 1.59 ^ab^
*B. bassiana* J18	25 ± 0.76	68 ± 2.37 ^bcd^	79 ± 2.90 ^ab^
*B. bassiana* FRh2	14 ± 0.37	14 ± 0.55 ^a^	68 ± 2.12 ^ab^
*B. pseudobassiana* FW Mana	25 ± 0.87	100 ± 3.76 ^e^	100 ± 4.46 ^b^
*B. pseudobassiana* I12 Damo	46 ± 2.87	100 ± 2.31 ^e^	100 ± 4.30 ^b^
*B. caledonica* F532	14 ± 0.28	25 ± 1.20 ^ab^	25 ± 1.30 ^a^
*B. bassiana* CTL20	46 ± 1.60	100 ± 4.02 ^e^	100 ± 3.05 ^b^
F2, 12	0.63	11.61	6.53
*p*-value	0.783	<0.001	0.001

Co-inoculum group A = TPP-H + FRh2 + F532; B = F615 + J2 + J18; C = FW Mana + I12 Damo + CTL20; D = F615 + I12 Damo + F532.

## Data Availability

Not applicable.

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
