# Peer review of "Biological Control of Diamondback Moth—Increased Efficacy with Mixtures of *Beauveria* Fungi"

_microorganisms, 2022, doi:10.3390/microorganisms10030646_

Round 1
Reviewer 1 Report
It is unclear that mixtures of Beauveria fungi show the increased efficacy. Please provide the graph which compare with negative or positive control. In addition, it needs to be provided the MoA (mode of action).
Reviewer 2 Report
Manuscript Title: Biological control of diamondback moth – increased efficacy with mixtures of Beauveria fungi
Authors: Sereyboth Soth, Travis R. Glare , John G. Hampton , Stuart D. Card , and Jenny J. Brookes
Manuscript ID: microorganisms-1630311
General comment:
In the manuscript, the authors evaluated the biological control efficacy of diamondback moth using mixtures of Beauveria fungi. This study found three isolates of B. bassiana and three of B. pseudobassiana are potential candidates for the control of diamondback moth. Further research on this area may result in the development of management strategies for biocontrol of diamondback moth.
In general, this paper is clearly laid out, well planed and easy to read. The experiments are well designed and appropriate controls are presented. Some specific suggestions or questions are listed below:
- Abstract: Authors need to revise the abstract for more accuracyand novelty.
- Introduction is easy to read but needs a little completed. For example, there are severalsynthetic insecticides used for effectively control diamondback moth. I suggest the authors include more information about these synthetic insecticides into this section and discuss why we need the biological control way.
- Introduction: Add references to support the statement “Biological control is an environmentally sound and effective means of reducing, or mitigating, insect pests in agriculture, aquatic, forest, natural resource, stored products, and urban environments.”
- The Introductionsection should focus on the research progress related to the topic and emphasize the innovation of this research. However, the novelty and significance of the topic were not highlighted, please modify the introduction more clearly.
- Results: Please include the Sequences of Table 1into text of the 2. DNA extraction, PCR and sequencing, or remove Table 1 as Supplementary Materials.
- Figures 3, 4 and 6: please use italics for the generic names.
- Please ensure that abbreviations/acronyms are defined the first time they appear in each of three sections: the abstract; the main text; the first figure or table.
- Manyof the references have been superceded and more modern ones are required, such as Journal of Economic Entomology, 1925. 18(2): p. 265-267; Mammalian Protein Metabolism, 1969. 3: p. 21-132; Payne, R., et al., Genstat 5 reference manual. 1987; Journal of Molecular Biology, 1990. 215(3): p. 403-410; Journal of Economic Entomology, 1998. 91(3): p. 624-630;
Reviewer 3 Report
The manuscript entitled "Biological control of diamondback moth – increased efficacy 2 with mixtures of Beauveria fungi" is focusing on testing potential of 14 different Beauveria spp. used as a biological control agents (as single strains or co-inocula) towards diamondback moth (DBM), an important pest of brassica crops.
The manuscript is well written. The introduction provides sufficient information, the materials and methods are also well described.
I have some editorial remarks, mainly about Figures and Tables (details attached in a pdf file).
I am also wondering if Authors would be able to slightly reduce the number of cited references. Now there are almost 110 items on the list. In many places Authors cite few articles in one time....

Round 2
Reviewer 1 Report
Thank you for your kind responses. I would like to suggest to accept in present form.
Reviewer 3 Report
Now the manuscript could be published.